# High Yield of Chest X-ray in the Follow-Up of Colorectal Cancer

**DOI:** 10.3390/jcm11133828

**Published:** 2022-07-01

**Authors:** Eline G. M. Steenhuis, Ivonne J. H. Schoenaker, Jan Willem B. De Groot, Jos A. Stigt, Onne Reerink, Wouter H. De Vos tot Nederveen Cappel, Henderik L. Van Westreenen, Richard M. Brohet

**Affiliations:** 1Department of Gastroenterology and Hepatology, Isala Zwolle, Dokter van Heesweg 2, 8025 AB Zwolle, The Netherlands; w.h.de.vos@isala.nl; 2Isala Oncology Center, Isala, Dokter van Heesweg 2, 8025 AB Zwolle, The Netherlands; i.j.h.schoenaker@isala.nl (I.J.H.S.); j.w.b.de.groot@isala.nl (J.W.B.D.G.); 3Department of Pulmonology, Isala, Dokter van Heesweg 2, 8025 AB Zwolle, The Netherlands; j.a.stigt@isala.nl; 4Department of Radiation Oncology, Isala, Dokter van Heesweg 2, 8025 AB Zwolle, The Netherlands; o.reerink@isala.nl; 5Department of Surgery, Isala, Dokter van Heesweg 2, 8025 AB Zwolle, The Netherlands; h.l.van.westreenen@isala.nl; 6Department of Epidemiology & Statistics, Isala, Dokter van Heesweg 2, 8025 AB Zwolle, The Netherlands; r.m.brohet@isala.nl

**Keywords:** colorectal cancer, colorectal carcinoma, follow-up, chest X-ray, chest radiograph

## Abstract

Purpose: Worldwide, colorectal carcinoma (CRC) has a high incidence and a substantial cancer-related mortality. The recurrence risk is 30–50% and lung metastases are common. Treatment of lung metastases with stereotactic ablative radiotherapy (SABR) or metastasectomy may increase survival. The best modality for thoracic screening in the follow-up, however, remains controversial. In this study, we aimed to unravel the additional value of routine chest X-ray (CXR) for detecting lung metastases during the follow-up of CRC patients treated with curative surgery. Methods: Between 2013 and 2017, 668 CRC patients were treated with curative intent, of whom 633 patients were included in follow-up, which consisted of CXR, serum Carcino-Embryonic Antigen (CEA) and ultrasound of the liver. Patients who developed lung metastases, diagnosed with CXR and characterised by a normal concomitant serum CEA level, were identified. Number, size and treatment of lung metastases were described. Results: Thirty-four (5.4%) patients developed lung metastases. Seventeen (50%) were detected by CXR without pathological CEA levels. Eleven (65%) of these patients were treated with curative intent, whereas 21% of patients with lung metastases and elevated CEA levels were treated with curative intent (*p* = 0.049). Higher numbers of lung metastases were associated with a lower chance of curative treatment. Conclusions: More than 50% of patients with lung metastases on CXR in the follow-up would not have been detected with CEA-triggered imaging only. In addition, patients with colorectal lung metastases without elevated CEA levels were often suitable for curative treatment and, therefore, CXR seems to have additional value within the follow-up of CRC.

## 1. Introduction

Colorectal carcinoma (CRC) is amongst the top three most commonly diagnosed cancers for both males and females. In 2012, the reported worldwide incidence was 1.4 million patients, with a mortality rate of almost 700 thousand patients [1]. After curative treatment, 30–50% of patients will develop recurrences, depending, particularly, upon the stage of their primary tumour [2]. Metastases are most frequently localised in the liver. The lungs, peritoneum and lymph nodes are common sites for extrahepatic spread [3]. Lung metastases are found in about 10% of patients [4,5]. Although extrahepatic spread is associated with poor prognosis [3,6], local treatment may improve survival [2,3,7]. Therefore, follow-up may need to focus on finding both liver and lung metastases.

Recent studies question the yield of follow-up regimes and the follow-up in general. Common follow-up strategies include imaging of the chest, ultrasound of the liver and measurement of serum Carcino-Embryonic Antigen (CEA) levels for up to 5 years after curative treatment [6,7,8,9]. Surveillance colonoscopies to screen for metachronous polyps or cancer are performed lifelong on a regular base. The COLOFOL trial [7] and ‘CEAwatch’ trial [8] are recent trials regarding this topic. The COLOFOL trial compared recurrence rate and mortality rate in patients with stage II and III colorectal cancer with intensive and less-intensive follow-up schemes. Although recurrences were detected earlier in the group with intensive follow-up, no significant differences in 5-year mortality or colorectal cancer-specific mortality were found. Authors of the ‘CEAwatch’ trial only used imaging when two subsequent rises in serum CEA levels were seen, leading to earlier detection of recurrences, more often in a curable stage. Mostly due to this trial, monitoring the tumour marker CEA has received great attention and a follow-up based on CEA-triggered imaging only is proposed.

In national and international CRC follow-up guidelines [6], the modality of choice for chest imaging remains controversial [6]. Chest imaging is recommended in stage II–IV CRC, but there is no recommendation on the best modality and frequency [6]. This is due to limited studies regarding this topic. One review [10] compared chest X-ray (CXR), Computed Tomography (CT) scans and Positron Emission Tomograpy (PET) scans as modalities for chest imaging in the follow-up of CRC. CT scans detected more lung metastases, but it remained unclear whether detection of these metastases had any clinical benefit. Another systematic review [11] described outcomes of two studies using CXR for detecting lung metastases. Follow-up with annual CXR resulted in only 0.6% of the patients in pulmonary metastasectomies, without survival benefit. These data are supported by the guidelines of the American Society of Colon and Rectal Surgeons [12], reporting that 1.8–12% of CXR in the follow-up of CRC revealed resectable lung metastases. 

It is obvious that CT is more sensitive than CXR in detecting colorectal lung metastases [10,13]. However, a small retrospective study comparing CXR and chest CT for follow-up could not demonstrate a difference in disease-free interval or complete resection rate [5]. Common practice of CRC follow-up amongst Dutch surgeons is CXR [14].

When lung metastases are found, different treatment options may improve survival. Five-year survival rates after pulmonary metastasectomy are reported to range from 21 to 61.4% [3,4,15]. About 50–75% of these patients will have recurrences again, both in the lungs and extrapulmonary. Several studies have reported high 5-year survival rates after the second and even third resection of lung metastases [3]. An alternative to surgery is high-dose stereotactic ablative radiation therapy (SABR), which is less intensive and spares lung tissue [16,17]. 

The primary goal in follow-up is detecting recurrent disease in a curable stage. Follow-up exams that identify recurrences in an asymptomatic stage may improve survival rate. Data on the value of CXR in addition to measuring CEA levels in the follow-up of CRC are scarce and studies are often of poor quality. Despite this, CXR is widely used as a surveillance test after curatively treated CRC. This study aimed to evaluate the additional value of CXR for diagnosing lung metastases in patients with CRC during follow-up.

## 2. Materials and Methods

### 2.1. Patients and Data Collection

Patients who were enrolled in follow-up after curative CRC surgery for stage I–IV disease in Isala Oncology Centre, Zwolle, The Netherlands, between January 2013 and December 2017 were included. Follow-up was conducted for up to 5 years and consisted of routine CXR, ultrasound of the liver and measurement of serum CEA. CXR was performed every 6 months in the first two years and yearly after that, according to protocol. Serum CEA levels were analysed in accordance with the ‘CEA-watch’ trial [8]. Consequently, when serum CEA was <2.5 μg/mL, a rise of ≥1 μg/mL was considered pathologic. When serum CEA was >2.5 μg/mL, a rise of ≥20% was considered pathologic. In patients with no earlier CEA measurements, serum CEA > 5.0 μg/mL was considered pathologic. When pathologic rise of serum CEA was detected, additional imaging with CT or PET scan was performed. Patients with pT1N0 disease, a second malignancy and/or patients who were simultaneously in follow-up schedules for different malignancies were excluded from analysis. In addition, patients without patient files in our current electronic patient file system were excluded. Not completing follow-up was no reason for exclusion. 

### 2.2. Statistical Analysis

Primary outcome of this study is the detection rate of lung metastases based upon an abnormal CXR, without pathological CEA levels. In addition, numerical and dimensional characteristics of the lung metastases are described. The timing of development of lung metastases found with different surveillance tests is demonstrated as well as the survival of patients with lung metastases detected by CXR compared to CEA. Lastly, the curatively intended local treatments are described. Baseline characteristics in the study group were collected and all CXRs performed in the context of follow-up of CRC were analysed. The CXRs were classified as suspect for metastatic disease or not suspect for metastatic disease. When metastases were suspected, a CT of chest and abdomen or PET scan was performed and subsequently patients were discussed during the multidisciplinary colorectal cancer meeting by gastroenterologists, surgeons, medical oncologists, pathologists, radiologists and nurse practitioners. In every patient with lung metastases, CEA levels were analysed. 

Descriptive statistics were used to summarise the variables. The variables included age, gender, localisation of the primary tumour and metastatic disease, disease stage and CT scan, CXR, CEA and treatment findings. We estimated the validity of the diagnostic modalities by calculating the sensitivity and specificity of the CXR. As, often when (small) pulmonary nodules were found, it was not always possible to obtain histopathological confirmation, the gold standard for lung metastases was the outcome of the multidisciplinary meeting, as is common practice in most hospitals. A chi-squares test was used to compare treatment intentions of patients with metastases found with different surveillance tests. The Mann–Whitney U test was used to calculate differences in treatment intentions between patients with different numbers of pulmonary metastases. Kaplan–Meier and Cox proportional hazards analyses were performed to evaluate the probability of survival after lung metastases. IBM Corp. Released 2016. IBM SPSS Statistics for Windows version 24.0. Armonk, NY, USA: IBM Corp was used for analysis. A *p*-value of less than 0.05 was considered statistically significant.

## 3. Results

Between January 2013 and December 2017, 668 patients with CRC were treated with curative intent. Of these, eight patients were lost to follow-up, five had a second malignancy and seventeen did not enrol in the follow-up or did not have CXRs. In four patients, the intention of the treatment changed to non-curative; hence, they had different follow-up schemes. One patient had a histological confirmed jejunum carcinoma, instead of CRC and was, therefore, excluded. In total, 633 patients (males = 360, 56.9%) were included in this study. Patient characteristics are reported in Table 1. Patients had a minimum of one and a maximum of ten CXRs (median five CXRs). Data on ethnic origin was not included in this study, due to its retrospective design and these data were not included in patient files by default.

As such, 34 out of 633 patients (5.4%) developed lung metastases during follow-up. Twenty-five of these patients had a CXR suspect for metastatic disease, of whom seventeen had normal CEA levels. Nine patients had normal CXRs, whilst having pulmonary metastases. Six of these patients were found due to increased CEA levels and three were found due to complaints or CT imaging because of other health problems. The sensitivity and specificity of CXR for finding pulmonary metastases were, respectively, 73.5% and 94.7% (Figure 1, Table 2).

Within the group of patients with lung metastases, 21 patients had colon cancer (5% of all colon cancer patients) and 13 patients had rectal cancer (7% of all rectal cancer patients). The timing of the development of lung metastases varied greatly, between 6 and 63 months after curative resection (median 20 months). In colon cancer, three (14%) patients developed lung metastases in the first year and, in rectal cancer, two (7.7%) patients. Most metastases were found in the second and third year after curative resection (Figure 2).

Fifteen out of thirty-four (44%) patients with lung metastases were treated with curative intent (Figure 2 and Figure 3). Patients with lung metastases found by an abnormal CXR had significantly higher chance of curative treatment than patients with increased levels of CEA (65% vs. 21%, *p* = 0.049, Figure 3). Out of 14 patients with lung metastases and elevated serum CEA levels, 6 patients had lung metastases solely. Of these patients, three were treated with curative intent. Out of 17 patients with lung metastases, found with an abnormal chest X-ray only, 12 had lung metastases solely (Table 1). Ten out of these patients were treated with curative intent. In the group of nine patients with lung metastases and normal CXRs, three patients were treated with curative intent (Figure 1). With an average of 37 months after finding the lung metastases, 13 (86.7%) of the curatively treated patients are still alive (Figure 3), 5 without recurrences. The median survival of patients with metastatic disease was 66% (95%CI = 39–71%) for normal CEA versus 48% (95%CI = 36–60%) for increased CEA levels. Although the hazard suggests improved chance of survival in normal CEA metastatic disease, this was not statistically significant (P_logrank_ = 0.33 and Hazard Ratio (HR)=0.63; 95%CI = 0.24–1.63; *p* = 0.340).

Table 3 shows the number and diameter of lung metastases per treatment strategy. In curatively treated patients, the number of metastases defines treatment type. Patients with one lung metastasis receive surgery, whereas patients with multiple metastases are treated with SABR. Furthermore, the diameter of the metastases did not influence treatment intention. 

## 4. Discussion

This study confirms the possible additional value of CXR in follow-up after curatively treated CRC because half of the patients with lung metastases had an abnormal CXR without elevated CEA levels. CXR had a sensitivity and specificity of, respectively, 73.5% and 94.7% in finding pulmonary metastases of CRC.

Both CEA and CT imaging have significant potential in finding pulmonary metastatic recurrences of CRC. The current literature explicitly emphasises the role of CEA in the follow-up of CRC and only ‘CEA-triggered’ imaging is proposed [8,18,19]. Studies on the benefit of CXR in the follow-up of CRC are scarce and a clear position is still debated. In this study, almost 60% of patients with lung-only metastases had normal CEA levels. Half of the patients with lung-only metastases had an abnormal CXR with normal levels of CEA, of whom two-thirds were treated with curative intent. CEA-triggered imaging, as proposed, would have missed these patients. 

These data show that patients with abnormal CXR and normal CEA have better chances of curative treatment. Survival analysis, although not significant due to low power, seems to be in favour of patients without elevated CEA levels. This is in accordance with the observation that patients with increased levels of serum CEA were more likely to have widespread metastases, instead of an exclusive pulmonary localisation. Studies regarding prognostic factors of patients undergoing pulmonary metastasectomies endorse the thought that high levels of CEA bear poorer prognosis [20,21,22]. This raises questions about a CEA-triggered-only strategy in the follow-up of CRC. The current findings may emphasise the additional value of imaging of the chest. Not only CEA level, but also number of lung metastases is of prognostic value [22]. This is reflected in our patients, who were more likely to be treated with palliative intention when having multiple lung metastases. 

The COLOFOL trial [7] showed that patients with stage II and III CRC with intensive follow-up regimes did not have improved survival over patients with less-intensive follow-up regimes. This might indicate that tumours that disseminate early do not have good prognosis. Most lung metastases in this study occurred in the second and third year after resection and less than 10% originated in the first year after resection. It is plausible that more metastases would have been detected earlier when follow-up was performed with CT scans. Although little is known about the biological behaviour of CRC lung metastases, our results suggest that patients with tumours disseminating early in the follow-up period are less frequently curatively treated. Although chest CT is superior to CXR in detecting metastases, the question remains whether finding metastases earlier results in a better survival [5,10]. The lower detection capacity of CXR compared to CT functions as a filter for larger, solitary and later-appearing metastatic disease.

In addition, CT has a higher likelihood of false-positive findings, leading to further investigation and more healthcare-related costs. In follow-up regimes using CT, indeterminate lung lesions were found in 4–42% of patients. After follow-up periods, more than 70% of these lesions proved of no clinical significance [10]. 

Limitations of our study are its retrospective design and a relatively small group of patients with pulmonary metastases. However, all CRC patients that were followed up after their surgery were included in the study, which makes potential selection bias less likely. Moreover, the 5.4% patients with lung metastases fell within a range of 1.8–12%, as was reported by the American Society of Colon and Rectal Surgeons [12]. Still, these data may feed the discussion on the optimal follow-up of colorectal cancer patients and support the need for further research. 

## 5. Conclusions

More than half of the patients’ lung metastases on CXR in the follow-up would not have been detected with CEA-triggered imaging, while these patients seem to have a higher chance of curative treatment. 

## Figures and Tables

**Figure 1 jcm-11-03828-f001:**
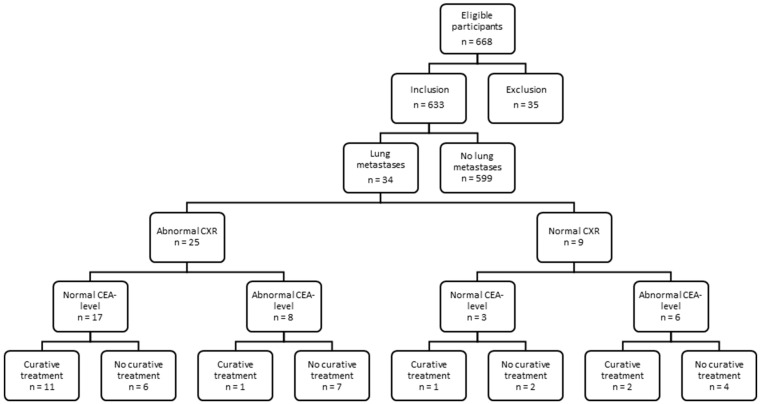
Flow chart of patients in the follow-up after curative treatment for colorectal cancer.

**Figure 2 jcm-11-03828-f002:**
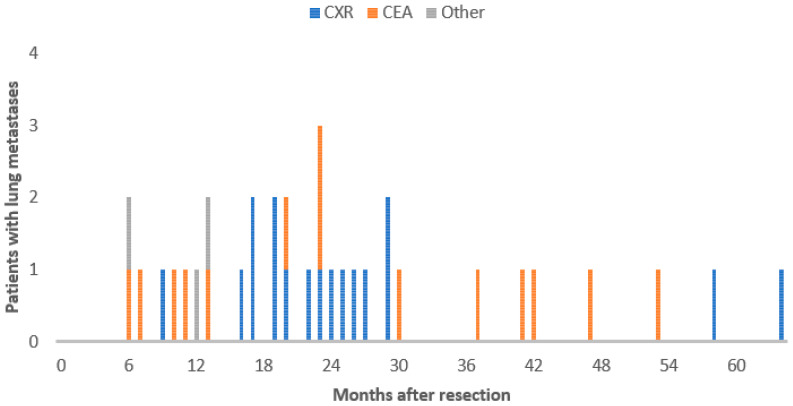
Timing of development of lung metastases found with different surveillance tests.

**Figure 3 jcm-11-03828-f003:**
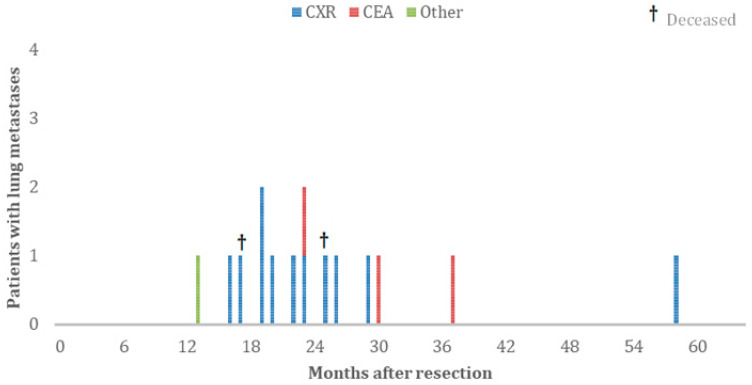
Patients with pulmonary metastases treated with curative intent, categorised by abnormal surveillance test.

**Table 1 jcm-11-03828-t001:** Patient characteristics.

Gender *n* (%)	
Male	360 (56.9)
Female	273 (43.1)
**Age**	
Average (sd)	67.7 (9.7)
**Localisation Primary Tumour n (%)**	
Colon	450 (71.1)
Rectum	178 (28.1)
Colon and rectum	5 (0.8)
**Disease Stage *n* (%)**	
I	176 (27.8)
II	253 (40)
III	191 (30.2)
IV	13 (2.1)
**Localisation of Metastases**	
**Pulmonary only**	**20**
CXR+, CEA−	12
CXR+/−, CEA+	6
CXR−, CEA−	2
**Pulmonary + other (i.e., liver, brain)**	**14**
CXR+, CEA−	5
CXR+/−, CEA+	8
CXR−, CEA−	1

**Table 2 jcm-11-03828-t002:** Chest X-rays in all patients and chest X-rays and CEA levels in patients with lung metastases.

		Lung Metastases		
		− *n* (%)	+ *n* (%)	Total *n* (%)
**Abnormal CXR**	**− *n* (%)**	567 (94.7)	9 (26.5)	576 (91)
	**+ *n* (%)**	32 (5.3)	25 (73.5)	57 (9)
**Total *n* (%)**		599 (100)	34 (100)	633 (100)
		**Abnormal CXR**		
		**− *n* (%)**	**+ *n* (%)**	**Total *n* (%)**
**Abnormal CEA**	**− *n* (%)**	3 (33.3)	17 (68)	20 (58.8)
	**+ *n* (%)**	6 (66.7)	8 (32)	14 (41.2)
**Total *n* (%)**		9 (100)	25 (100)	34 (100)

**Table 3 jcm-11-03828-t003:** Numerical and dimensional characteristics of the lung metastases.

Treatment Strategy	Number of Metastases		Diameter of Metastases	
	Range (*n*)	Median (*n*)	Range (mm)	Average (mm)
**Conservative**	1–9	3	10–59	21.5
**Palliative chemotherapy**	1–32	5	9–32	16.6
**Stereotactic ablative radiotherapy**	1–4	2	4–25	14.8
**Video-assisted thoracic surgery**	1	1	9.4–35	17.4

## Data Availability

Data available on request.

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
