# Peer review of "High Yield of Chest X-ray in the Follow-Up of Colorectal Cancer"

_jcm, 2022, doi:10.3390/jcm11133828_

Round 1

Reviewer 1 Report

The manuscript entitled “High yield of chest X-ray in the follow-up of colorectal cancer” by Steenhuis, E.  et al. evaluated 633 CRC patients followed-up for metastasis by CXR, serum Carcino-Embryonic Antigen (CEA) and ultrasound of the liver between 2013 and 2017 in Isala Oncology Centre, Zwolle, The Netherlands.

Although controversial, the authors evaluate the addition of CXR in the routine for detection of lung metastasis of CRC follow-up of patient submitted to curative surgery treatment.

The authors conclude that more than half of the patients with lung metastases on CXR in the follow up would not have been detected with CEA-triggered imaging while these patients seem to have a higher chance of curative treatment.

The results are interesting. To the final version the correction of table 1 misconfiguration is needed.

Author Response

The manuscript entitled “High yield of chest X-ray in the follow-up of colorectal cancer” by Steenhuis, E.  et al. evaluated 633 CRC patients followed-up for metastasis by CXR, serum Carcino-Embryonic Antigen (CEA) and ultrasound of the liver between 2013 and 2017 in Isala Oncology Centre, Zwolle, The Netherlands.

Although controversial, the authors evaluate the addition of CXR in the routine for detection of lung metastasis of CRC follow-up of patient submitted to curative surgery treatment.

The authors conclude that more than half of the patients with lung metastases on CXR in the follow up would not have been detected with CEA-triggered imaging while these patients seem to have a higher chance of curative treatment.

The results are interesting. To the final version the correction of table 1 misconfiguration is needed.

Answer: Thank you for the comments; we have corrected the misconfiguration of table 1.

Changes to the manuscript: We moved table 1 in the manuscript.

Reviewer 2 Report

The manuscript entitled "High yield of chest X-ray in the follow-up of colorectal cancer", is a good addition in field of dignostic sciences. The write up, results and conclusion seems fine. 

The manuscript entitled "High yield of chest X-ray in the follow-up of colorectal cancer", is a good addition in field of diagnostic sciences. However, the only comparison being done is between CXR and CEA. CEA itself is never been the accurate marker for detection of cancer and several condition for example during inflammatory states the CEA is found to be raised.

Secondly, the author themselves mentioned during discussion that CT is preferable scan for metastatic detection. In this instance, in my opinion the chest x-ray, can’t be a diagnostic method in sensitive cases of metastasis. However, this article could be an addition to research but not any innovative or advanced idea.

Author Response

The manuscript entitled "High yield of chest X-ray in the follow-up of colorectal cancer", is a good addition in field of dignostic sciences. The write up, results and conclusion seems fine. 

The manuscript entitled "High yield of chest X-ray in the follow-up of colorectal cancer", is a good addition in field of diagnostic sciences. However, the only comparison being done is between CXR and CEA. CEA itself is never been the accurate marker for detection of cancer and several condition for example during inflammatory states the CEA is found to be raised.

Answer: We thank the reviewer for this comment. Also, we agree with this statement; indeed, there are other circumstances than cancer recurrence that may influence CEA levels. However, the CEA-watch trial1 demonstrated a follow-up with CEA-triggered imaging only that led to earlier detection of colorectal carcinoma recurrences, more often in a curable stage. In accordance to this trial, the new Dutch colorectal carcinoma guideline advises to monitor CEA levels every 3-6 months in the first two years and every 6-12 months in the following three years. Twelve months after treatment with curative intention, a CT-thorax/abdomen is advised. Other imaging is only required when CEA levels are inappropriate, according to the CEA-watch trial. Many other international guidelines propose CEA-triggered imaging as well.

Changes to the manuscript: no changes were made according to this comment.

Secondly, the author themselves mentioned during discussion that CT is preferable scan for metastatic detection. In this instance, in my opinion the chest x-ray, can’t be a diagnostic method in sensitive cases of metastasis. However, this article could be an addition to research but not any innovative or advanced idea.

Answer: We thank the reviewer for this comment. International guidelines remain indecisive about recommendations regarding imaging of the thorax. They agree that imaging in necessary, but fail to give clear advice on the preferred modality of imaging. Since chest X-ray is common practice in many hospitals, we tried to unravel its diagnostic value in the follow-up of colorectal carcinoma. We agree with the comment of this reviewer that this article is an addition to research, although not innovative. Our data may feed the discussion on the optimal follow-up of colorectal cancer patients and support the need for further research.

Changes to the manuscript: No changes were made.

Reviewer 3 Report

This study presents compelling evidence for including CXRs in monitoring patients that underwent curative colorectal treatment. The usage of CXRs every 6 months for 2 years then annually thereafter, showed that more than half of patients with lung metastases detected by CXR did not have elevated CEA levels. Furthermore, the addition of using CXRs as part of the follow-up protocol resulted in better rates for curative salvage therapies.

Limitations of the study include the definition of pulmonary metastatic disease determined ultimately by the consensus of the multidisciplinary meetings (although understandable) can introduce false positive cases without usage of histopathological confirmation (or PET modality despite inherent limitations such as lesion size). As usual, reliability of results is not optimal due to the retrospective nature of the study and the limited sample size.

Review of the current follow-up practice for CRC patients and the impact of CXRs and CEA levels were practical and sufficient without being unnecessarily lengthy.

Presentation of study methology and results were clear and concise.

This manuscript is scientifically sound with an appropriate study design to address the benefit of including CXRs as part of the follow-up protocol. Ethics for the study was appropriate.

The study conclusions are supported by the presented results. The comparison between follow-up with CXRs vs. CT studies was concise and effective.

Overall, the study adds good value to further discussions/larger investigations for including routine CXRs as a part of the surveillance protocol for patients that underwent curative colorectal treatment.

Author Response

This study presents compelling evidence for including CXRs in monitoring patients that underwent curative colorectal treatment. The usage of CXRs every 6 months for 2 years then annually thereafter, showed that more than half of patients with lung metastases detected by CXR did not have elevated CEA levels. Furthermore, the addition of using CXRs as part of the follow-up protocol resulted in better rates for curative salvage therapies.

Limitations of the study include the definition of pulmonary metastatic disease determined ultimately by the consensus of the multidisciplinary meetings (although understandable) can introduce false positive cases without usage of histopathological confirmation (or PET modality despite inherent limitations such as lesion size). As usual, reliability of results is not optimal due to the retrospective nature of the study and the limited sample size.

Review of the current follow-up practice for CRC patients and the impact of CXRs and CEA levels were practical and sufficient without being unnecessarily lengthy.

Presentation of study methology and results were clear and concise.

This manuscript is scientifically sound with an appropriate study design to address the benefit of including CXRs as part of the follow-up protocol. Ethics for the study was appropriate.

The study conclusions are supported by the presented results. The comparison between follow-up with CXRs vs. CT studies was concise and effective.

Overall, the study adds good value to further discussions/larger investigations for including routine CXRs as a part of the surveillance protocol for patients that underwent curative colorectal treatment.

Answer: We thank the reviewer for these comments.

Changes to the manuscript: no changes were made.